# Addressing the Microplastic Dilemma in Soil and Sediment with Focus on Biochar-Based Remediation Techniques: Review

Heba Elbasiouny [1,*] and Fathy Elbehiry [2]

1   Faculty of Agriculture (Girls), Al-Azhar University, Cairo 11651, Egypt
2   Basic and Applied Agricultural Sciences Department, Higher Institute for Agricultural Co-Operation, Shubra El-Kheima 13766, Egypt; fathyelbehiry@gmail.com
*   Correspondence: hebaelbasiouny@azhar.edu.eg or hebaelbasiouny1@gmail.com

**Abstract:** Microplastic (MP) pollution is a widespread global environmental concern, representing an emerging contaminant with major implications for ecosystems and human well-being. While extensive research has focused on MPs in aquatic environments, their impact on sediments and soils remains inadequately explored. Studies have confirmed the harmful effects of MPs on soil and sediment biota, as well as on the properties of these ecosystems. Furthermore, the long-term persistence of MPs within the environment contributes to their accumulation in terrestrial and marine food chains, with potential consequences for groundwater quality. Although several methods have been applied to mitigate MP pollution, some methods have drawbacks and some are not studied well, necessitating the urgent exploration of novel, sustainable, and eco-friendly approaches. Biochar is a remarkable solution for pollution removal; recently it has been used in addressing the increasingly concerning issue of microplastic contamination. This review aims to shed light on the difficulty posed by MPs in soils and sediments, while highlighting the remediation methods and the potential advantages of utilizing BC as an environmentally friendly solution for MP removal and remediation.

**Keywords:** microplastic pollution; soil; sediment; traditional remediation of microplastic; biochar

## 1. Introduction

Recently, the potential negative impacts of emerging contaminants (ECs) have globally received massive attention over environmental and health concerns [1–4]. Anthropogenic activities are recently considered as the primary drivers of environmental degradation [5]. The widespread distribution of MPs in natural ecosystems originates from different activities, either land-based or marine-based, and MPs are extensively distributed in freshwater, marine, and terrestrial ecosystems [6,7]. The United Nations Environment Programme recently listed MP pollution as one of the top ten environmental problems. As a result, plastic pollution is increasingly regarded as a major contributor to the worldwide decrease in biodiversity, as well as a severe danger to the functioning of the Earth's systems and human health [1,8].

Every year, 381 million tons of plastic waste are produced globally (only 5–9% of that has been recycled). Of that, 38 million tons are generated by the US alone [9,10], and 15 and 32% of global plastic production in 2020 was accounted for by Europe and China, respectively [11]. Shopping bags are robust, inexpensive, and lightweight plastic goods that are frequently utilized on a worldwide scale. Shopping bags are made from nonrenewable resources such as petroleum and natural gas. They were first introduced in the 1970s, but nowadays, about 500 billion plastic bags are used globally every year [12]. Palansooriya et al. [13] reported that by 2050, it is predicted that 12,000 Mt of plastic trash would have entered natural ecosystems due to current plastic manufacturing patterns and inadequate waste management techniques. Therefore, a better understanding of the effect of MPs on soil ecosystems is required.

The three most often used plastic polymers in packaging are polypropylene, low-density polyethylene, and high-density polyethylene [14]. After prolonged exposure to sunlight and atmospheric factors, plastic products, especially bags, become fragile and readily fragment into small sizes, producing persistent particles, usually referred to as MPs [15]. Thus, the plastic waste may comprise micro- and macro-plastic; the size of MPs is <5 mm, mainly fragments, granules, fibers, films, foam, etc. [6,16,17]. O'Kelly et al. [8] stated that MPs can be degraded into smaller-sized plastics, like nano-plastics (NPs), or totally transformed into carbon dioxide and water in soil ecosystems. Because of their incredible endurance and durability and resistance, they are practically considered indestructible in specific environmental circumstances, such as those prevailing in the ocean [18]. However, plastic waste, including biodegradable plastic, is more susceptible to physical fragmentation than degradation, resulting in smaller plastic sizes. Due to the combined action of physical, chemical, and biological factors, plastic waste exposed to the natural environment will undergo weathering processes, such as decomposition and degradation [5]. Lee et al. [19] added that environmental weathering, by chemical oxidation and photo-oxidation by UV, causes the breakdown of plastics into small parts. Thus, MPs are classified into two types based on their origin (Figure 1): primary MPs that are generated as pellets, that is, micro-sized particles from manufactured plastics and domestic products such as cosmetics and personal care products, and secondary MPs, which are derived from the degradation of large-sized plastics or the biological, chemical, or physical weathering of plastics into forms such as fibers, foams, fragments, or films [4,6,16,20].

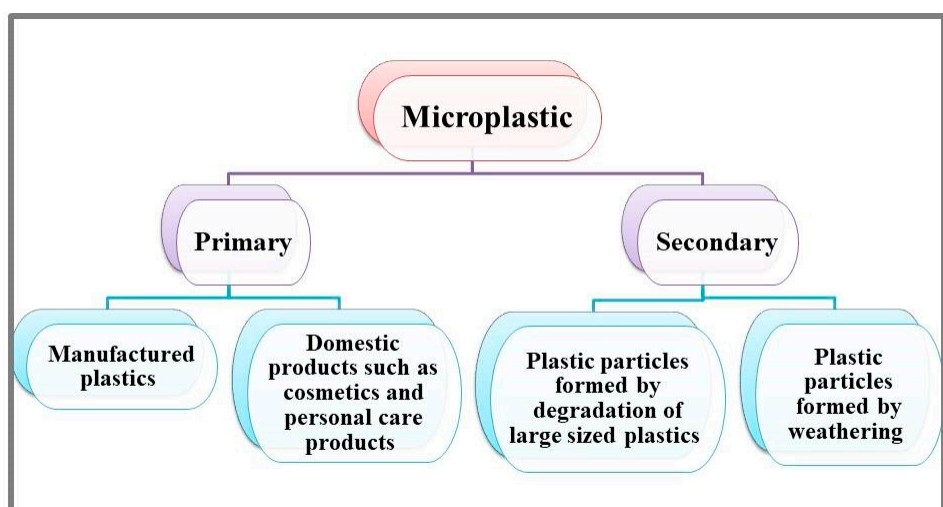

**Figure 1.** Primary and secondary microplastics.

Most of the work about MPs has been performed on aquatic ecosystems, particularly oceans, or even on sediments, beaches, and sludges [21]. Worldwide reports of MPs in the water column and marine sediments have been documented [22]. As a result, there is a knowledge gap regarding MP contamination in agricultural soil and terrestrial ecosystems [21]. Biofouling by prokaryotes, eukaryotes, and invertebrates can increase the density, size, form, and fluid density of MPs, causing them to sink to the bottom sediments. It has been noted that sediments are important MP sinks, and thus an accumulation of MPs in sediments can be harmful to both marine and human life [22–24]. A growing body of research suggests that MP particles and films can be digested by wildlife and hence enter the food chain, resulting in larger environmental and health effects of MPs than previously. These negative effects may be attributed to the oxidative stress caused by MPs in addition to the ability of MPs to selectively adsorb xenobiotics and other chemical compounds, particularly those with poor water solubility [15,25]. Due to their unique surface structures and characteristics, MPs have the capacity to absorb heavy metals and organic contaminants from soil solution and concentrate them locally in the soil [26]. As

well, MP accumulation in the environment can result in various problems and potential loss of ecosystem services [27]. Increasing information suggests that MPs can also impact the soil–plant system and they are more likely to accumulate throughout the soil food chain. It is reported that MPs affect biological, chemical, and physical soil characteristics, including its evaporation and water-holding capacity, porosity, aggregate size and formation, pH, and nutrient availability. However, the research of MP pollution is still in its early stages, particularly in soil environments [26]. Thus, there was and still remains a knowledge gap in this area of research [16,28–30]. Several methods, either physical, chemical, or biological methods, are used to remove MPs from the environment. Biochar (BC) has been considered in MP remediation; it basically acts as a filter that separates MP particles by trapping, adsorbing, and entangling these particles on its surface. It is a cost-effective and eco-friendly method of removing MPs [31]. However, there is a lack of reviews about MP removal and remediation by BC (or its modified forms such as activated, mineral-oxidated, magnetic, or nano), as well as its mechanisms and effectiveness. Therefore, this review highlights some objectives, such as addressing the MP dilemma in soil and sediment and its fate in the environment, as well as focusing on the use of BC in MP removal from the environment to highlight its potential advantages as an eco-friendly method in this widespread issue.

## 2. Microplastic in Soil

MP pollution in soil has attracted minimal scientific attention compared to that in the marine environments, despite the fact that it has recently been documented that soil is a major sink of MPs (4–23 times larger in its mass than marine environments) [5,25,32,33]. Microplastics have been found in soils all over the world, including Asia, Europe, North America, Africa, and Oceania, with abundances ranging from 870 particles $kg^{-1}$ in home garden soil to 42,960 particles $kg^{-1}$ in farmed regions. The great majority of MPs in soil were polyethylene (PE) and polypropylene (PP) [34]. However, the potential impacts of MPs in terrestrial environments are largely unexplored, despite the fact that more than 80% of plastic pollution in the oceans is usually produced, used, and disposed of on land [33]. Thus, MPs tend to accumulate more in terrestrial soil than in aquatic environments. According to the United Nations Environment Programme (UNEP), huge amounts of particulate plastics detected in the marine ecosystem worldwide are generated from land-based sources. Annually, 4.8 to 12.7 Mt of terrestrial plastic waste reaches the ocean, accounting for 1.7–4.6% of total plastic waste created. Sediment transfer through soil erosion is a process that enables plastic particulates to be transported from land to aquatic habitats. Despite this connection to land sources, most scientific studies on plastic particulates have ignored their consequences [5].

### 2.1. Sources of Microplastics in Soil

The sources of MPs in soil include landfills, inorganic fertilizers and organic waste application (such as sewage sludge and compost) that are contaminated by plastics, irrigation by wastewater, irrigation pipes, wastewater treatment plants, agricultural usage of plastic films (such as plastic mulch (broadly used to improve water use efficiency and plant growth) and greenhouse coverings), packaging bags, plastic-coated fertilizers, fertilizer/seed containers, textile applications, industrial waste, road dust, and dispersed atmospheric depositions of fibers and fragments [1,3,25,32,33,35–39]. As well, irrigation water and other diffuse sources can all introduce plastic into agricultural soils [35]. In addition, some products of personal care, abrasive industrial resin tablets, fragmented plastics produced through photolysis, abrasion, and biodegradation, and treated wastewater from the textile processing industry are other sources of MPs [40]. Agricultural films and compost applications are the most likely important sources of MPs in soil among the previous sources [1,41]. Thus, generally, the source of MPs in terrestrial environments can be classified into two groups: (1) point sources (such as plastic factories, landfills, and

wastewater treatment plants) and (2) non-point sources (such as roads and agricultural lands) [42] (Figure 2).

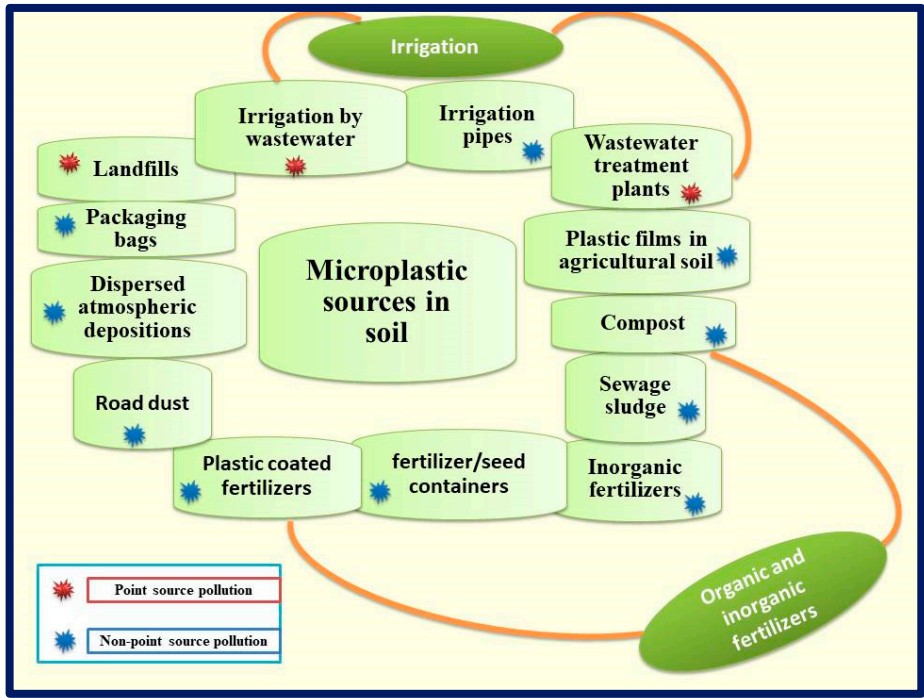

**Figure 2.** Microplastic sources in soil. Red star refers to the source of contamination by microplastics in soil as point source pollution (i.e., concentrated) and the blue one refers to non-point source pollution (not concentrated).

### 2.2. Risks of Microplastics in Soil

It has been stated that soil has been polluted with plastic to concentrations up to 7%. However, some of the recent reviews on MPs in soil have revealed that little is understood about the plastic effects in soil systems [35,43,44]. As a result, MP pollution and its hazards in terrestrial ecosystems have received much attention in the past few years [30,34].

2.2.1. The Impact on Soil Ecosystem Services, Soil Properties and Functions

The persistence and limited biodegradability of MPs in the soil ecosystem is one of the reasons that has attracted the interest of ecologists and environmental scientists worldwide. Furthermore, easy transfer of MPs to the soil environment from various sources poses a threat to the soil ecosystem [45]. Because MPs are abundant in many habitats, they are readily accessible to a wide range of organisms, including plants (a vital component of the terrestrial ecology), animals, and, eventually, humans [30]. It is confirmed that MPs can also exert certain eco-toxicological effects on the physicochemical, faunal, and microbiological properties of the soil and its biodiversity [17,45]. Once accumulated in the soil, MPs also directly or indirectly affect the soil ecosystem functions. The MPs have been noticed to cause soil health deterioration by negative effects on the soil bulk density, soil porosity, water-holding capacity, aggregates, and soil structure. In addition, it slows nutrient cycles and has harmful effects on soil organisms [21,43]. Moreover, it is argued that MPs could be a part of soil C storage, which complicates the mixture of soil organic matter by simplifying the formation of high-molecular-weight aromatic compounds [43]. Also, it was shown in research on the impact of MPs and biodegradable mulch on soil chemistry that the examined MPs increased soil pH through interactions with different cations and protons in both organic and inorganic components. This demonstrates how these MPs change pH in various soil types, reducing pH in acidic soils and raising pH in alkaline soils, for example.

For a wide variety of soil and MP types, further field and lab investigations are advised because it is unknown what processes are responsible for these alterations [46].

Upon reaching the soil, MPs may have an impact on the stability and operation of soil ecosystems because of the differing characteristics of MPs and soil components (such as soil minerals and soil organic matter). For instance, because of their lower density, MPs often diminish soil bulk density. MPs are a form of solid pollutant that may embed inside soil aggregates and alter the structure of the soil. According to several research studies, MPs may cause a decrease in soil aggregates [47]. According to Li et al. [47], this is explained by the possibility that MPs might cause fracture spots in soil aggregates. In contrast, Zhang et al. [48] discovered that in the pot experiment, numerous wet–dry cycles increased the production of macro-aggregates (>2 mm), while this was not proved in the field experiment. Changes in bulk density and soil structure had an impact on soil porosity, which in turn had an impact on soil permeability and water-holding and supplying capability. Additionally, MPs had an impact on several soil chemical characteristics, including pH, electrical conductivity, organic carbon content, and nutritional components [31,47]. MPs may also have impacts on soil biological indicators such soil enzyme activity in addition to their influence on soil's physicochemical attributes. For instance, the fluorescein diacetate hydrolase activity was lowered by PE and PVC MPs, whereas the urease and acid phosphatase activities were elevated [46].

### 2.2.2. Association with Other Pollutants

Microplastics have been labeled as Trojan horses (vectors) since they can absorb and transmit other contaminants while also releasing toxic organic compounds [49]. Due to the unique characteristics of MPs, they have a remarkable adsorption capacity for several contaminants [3,14] via H bonding, ion complexation, and electrostatic interaction [25]. The MPs have an affinity for some hazardous hydrophobic organic chemicals, persistent organic contaminants, and non-essential trace elements (such as polychlorinated biphenyls, additives, dichlorodiphenyltrichloroethane, plasticizers, and heavy metals) [14]. Thus, contaminant adsorption by MPs enables MPs to behave as contaminant carriers or drivers, complicating the research of MPs' eco-toxicology [50]. Evidence suggests that MPs can serve as a vector for the transmission of pollutants (such as heavy metals, hydrophobic organic pollutants, pharmaceuticals, personal care products, and even human pathogens) absorbed from soil, including plastic additives and some other pollutants. In addition, it affects the transport of pollutants, posing potential dangers to soil biota [1,34,51,52].

Yao et al. [53] reported that colloidal particles in porous materials are known for their ability to aid or hinder the transport of heavy metals. Soil MPs, like other colloidal particles, can react with metals due to their highly reactive functional groups (e.g., $CH_3$-, $NH_2$-, $COO$-, $-C-C-$, and $-SO_3H$) and large specific surface areas. Plastic may cause synergistic pollution with heavy metals, posing a risk to ecosystems. Joint exposure to MPs and cadmium, for instance, may result in varying degrees of toxicity to plant development. As well, organisms that consume MPs that have toxic chemicals adsorbed to them may face significant health concerns since these chemicals can be transmitted into their organs and tissues. It has been established that MPs can act as carriers for harmful substances (such as polychlorinated biphenyls, nonylphenols, and polyaromatic hydrocarbons) into organisms [54].

As the surface of plastic is negatively charged, it reacts with positively charged soil particles or ions, forming a complex combination of minerals and organic matter. As a result of this interaction, it can influence dissolved organic matter, extractable ions, bulk density, water-holding capacity, and aggregate stability, which affects the soil physical and chemical environment, soil (micro)organism habitat, and consequently plant development [35]. The interactions between MPs and pollutants are unavoidable when they coexist, and these interactions can influence the toxicity of particular pollutants. A small number of research studies have shown that hydrophobic organic pollutants can affect MPs' ecotoxicity. The adsorption capability of triclosan on MPs, for example, can have a direct impact on the toxicity of various MPs. It is reported also that MPs can boost the effects of an organophosphorus

flame retardant on mice to promote oxidative stress, amino acid metabolism, neurotoxicity, and energy metabolism [55]. The Cd adsorption by soil was reduced by the presence of high-density polyethylene MPs, while Cd desorption from soil was increased. Plant performance and soil fungi are influenced directly and indirectly by the co-occurrence of MPs, which prompts changes in Cd availability and soil characteristics [3]. As well, Li et al. [45] reported that MPs, due to their high surface area, may bind hydrophobic compounds like PAHs effectively, which raises their risk when taken by biota. As a result, the impacts of combined pollution from MPs and other pollutants on terrestrial species should be more closely considered [34].

### 2.2.3. The Impact of Microplastics on Soil Biota, Microbes, and Food Chain

The presence of MPs has an impact on soil microbes, resulting in alterations in both the diversity of the microbial community and the activities of enzymes within the soil ecosystem. This affects the overall health of the soil ecosystem. However, the influence of MPs on soil microbiota and associated biogeochemical processes is not well-understood. Despite the increasing attention given to this phenomenon, the underlying mechanism remains unknown. Therefore, it is necessary to conduct further research to investigate the dosage dependence and specific types of MPs that affect soil microbial communities and enzyme activities [45]. Ma et al. [55] stated that MPs may present an emerging danger to terrestrial ecosystems by influencing soil biota at various trophic levels. MPs can also be ingested by terrestrial organisms such as ciliates, flagellates, amoebae, collembolans, and earthworms, which causes a decrease in their survival and growth rate, digestive damage, disruption of immunity, neurotoxicity, oxidative stress, abnormal gene expression, DNA damage; they can even be transferred up the food chain (Figure 3) [36]. After ingestion, MPs pass on to bigger soil species, resulting in unknown consequences for soil fauna and microbes. Several species were more widespread in MP-contaminated soil, including plastic-degrading bacteria and pathogens, revealing that MPs can work as a unique microbial habitat, possibly changing the biological processes of soil ecosystems [33].

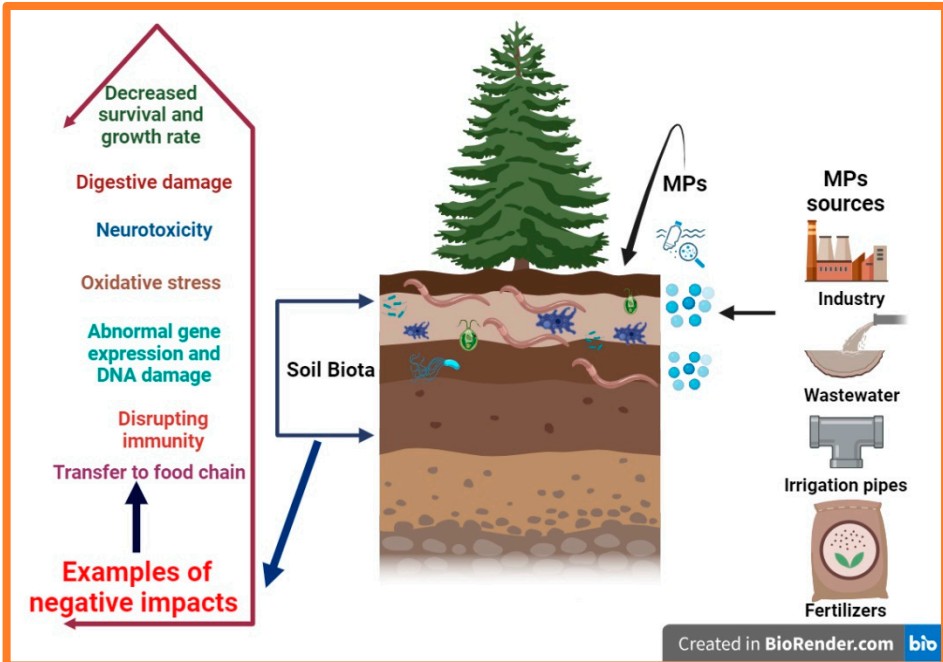

**Figure 3.** Transferring microplastic through soil, affecting soil biota and food chain.

Since the diversity and relationships between soil microorganisms are important causative factors in soil aggregation, soil exposure to MPs can alter the relationship between microbial activity and water-stable aggregates. Soil microorganisms are the essential pool

of soil living biomass and they play key roles in biogeochemical cycles. As a result, C and N biogeochemical cycling may vary if microbial populations are significantly changed by MPs, influencing soil ecosystem functions and services [43].

Although the impacts of MP pollution on soil macro-organisms have gained a lot of attention, research on their impacts on the microbial community is uncommon, especially when it comes to plant–soil–microbe interactions and its significant impacts, such as nutrient cycling, soil organic matter, C storage, pollutant attenuation, and total $CO_2$ flux [33].

### 2.2.4. The Impact of Microplastics on Ground Water

Groundwater is one of the most significant natural resources on the planet. Groundwater accounts for 1.69% of total water accessible on Earth, with just 0.76% of fresh and usable water. Despite its scarcity, groundwater contributes significantly to many nations' daily water requirements [56]. Groundwater is a key source of water for many purposes, such as drinking, agriculture, residential, and industry, for over two billion people [37,57]. The urbanization and population growth of rural regions puts pressures on and depletes groundwater systems, which reduces groundwater quality. Microplastics, among other emerging contaminants, became highly concerning, owing to their persistence in the environment [57]. A substantial amount of plastic garbage is dumped into landfills and thrown in rivers and other surface water bodies and the ocean, where it degrades into smaller fragments due to the action of biotic and abiotic processes and eventually finds its way into groundwater. However, there is little known regarding the presence of MPs in the groundwater [56].

Microplastics in soil can also be a source of MPs in groundwater. Seepage via pores and cracks, in addition to interactions with colloidal aggregates, can have effects on the dynamics of MPs in the subsurface soil, which makes the identification of MPs in groundwater systems problematic. Microplastics gradually migrate from the topsoil to deeper layers of the soil, where they accumulate and ultimately move to other areas of the ecosystem with the aid of water and animals. The risk of loss of MPs is estimated to be much greater, especially in artificial drainage agricultural soils, huge quantities of macro-pores, and with occurring surface runoff [1,53,57,58]. Furthermore, the combinations of mechanical corrosion, UV light, weathering, and (micro)biological degradation would break plastics into small particles and disintegrate plastic waste in natural aquatic and soil systems at the micro-to-nano scale [53,59]. The health risks that are posed by MPs are induced in three ways: (1) hazards caused by MP particles through ingestion; (2) hazards caused by different chemicals, colors, plasticizers, and ingredients present in the MPs; and (3) hazards caused by the creation of biofilm by micro-organisms over MP particles. Furthermore, when introduced into the aquifer system, MP pollution poses serious risks to groundwater ecology and the soil environment, ranking it among the top 10 environmental issues [56].

Microplastics have the potential to contaminate groundwater through various contamination pathways. This contamination is influenced by hydrogeological factors such as the source of groundwater recharge (such as the vadose zone or surface water) and the timing of recharge or recession periods. The source of MPs can be diffuse (like a losing river) or point-specific (such as leaks from the drainage system). The characteristics of the aquifer, such as hydraulic conductivity, also play a role in the percolation and transport of MPs. Additionally, the interaction between surface water and groundwater further affects the movement of MPs. Unlike solute contaminants, the movement of MPs in the aquifer is also influenced by the size of MP pores and dimensions [60]. Khant and Kim [37] stated that even though groundwater contamination can have an impact on human health, plant varieties, and subsurface micro-organisms, there have been far less investigations on MPs in groundwater compared to soil. Thus, the presence of MPs in groundwater should not be ignored. It requires immediate focus from the scientific community, particularly hydrogeology and environmental impact research, to reduce their adverse effects and evaluate the potential risks on human society and the environment.

### 3. Microplastics in Sediments

From an ecological perspective, sediments are regarded as the primary and most important component of water systems and have been identified as a location for the deposition of a variety of contaminants, including MPs [61]. The worldwide distribution of MPs in the marine environment is well-established based on the information presented thus far. Their widespread presence in the marine ecosystem causes significant levels of contact with the biota in sediments, deep abysses, and surface waters [62]. In the aquatic ecosystem, MPs can float or sink to the bottom based on the polymer composition. Low-density polymers like polypropylene and polyethylene can float on the surface; thus, these polymers stay on the surface, whereas high-density polymers such as PVC, polyamide, and polyester sink to the bottom (Figure 4) [52]. It is indicated the MP distribution in the sediment is higher than the surface of the water bodies (approximately up to 5 times) [63]. Plastics can undergo density modification once they are in the water by processes such as additive leaching, biofouling, and assimilation within marine aggregates. Even if their initial densities kept them floating, these processes make it easier for MPs to sink to the ocean floor. The degradation of plastic trash may then be slowed by the low-energy environment, low oxygen levels, freezing temperatures, and lack of solar UV light in the benthic zone. This could make MPs in the marine environment more persistent [64]. Thus, sediments, like soils, serve as a long-term sink for MPs in deep-sea and river-estuary areas, which either allow them to be adsorbed on their surfaces or be trapped between their spaces [25,65]. Denser polymers sink and tend to accumulate in the sediment at the bottom. The disruption caused by wind conditions affects the dispersion of MPs in the water column. Thermal, chemical, hydrolysis, mechanical degradation, and photodegradation of MPs can also reduce the particle size at the bottom [52]. Synthetic polymers are found in lakes, rivers, and seas across the world and accumulate in sediments because of their properties, which include resistance and high durability. Tiny particles are readily ingested by a variety of aquatic species, accumulating in their cells and tissues until being transferred via the food chain (Figure 5). Thus, most plastics are highly robust and long-lasting [14].

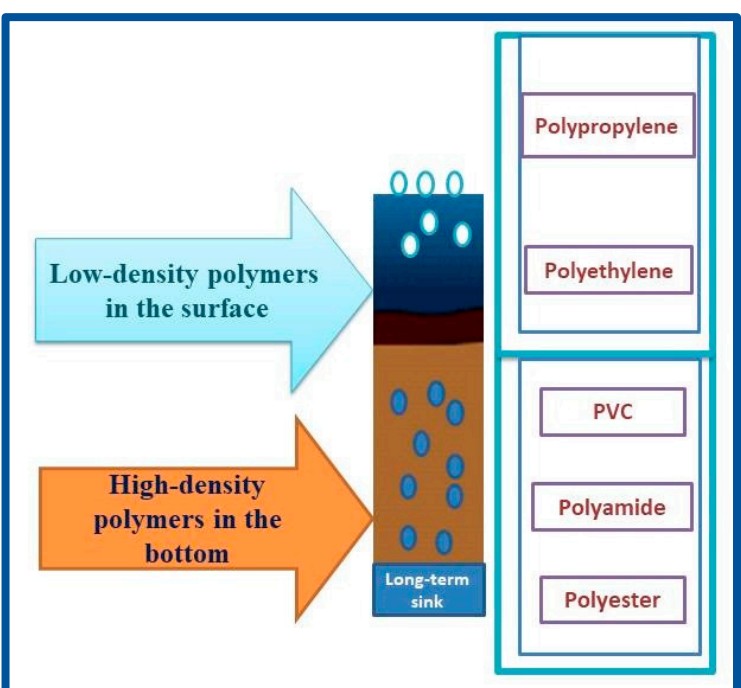

**Figure 4.** Distribution of microplastics in the aquatic ecosystem.

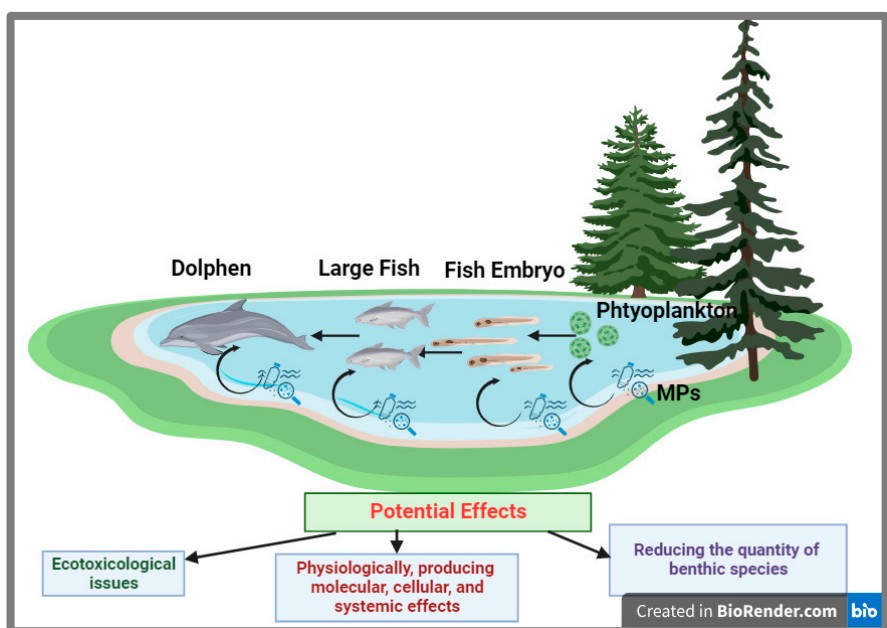

**Figure 5.** Transfer of microplastics (MPs) through aquatic food chain and its effects.

Given that many sediment organisms are filter or detrital feeders and are thus more vulnerable to swallowing MPs, the widespread presence of MPs in sediments raises possible eco-toxicological issues. Once consumed, MPs can harm aquatic species physiologically, producing molecular, cellular, and systemic effects as well as apical endpoint damage and hepatic stress. In addition, up to 90% of the biomass of fish food comprises benthic invertebrates, so exposure to MPs in sediment may result in its bioaccumulation and trophic transmission. Exposure to MPs may completely reduce the quantity of macro-invertebrates and other benthic species at the community level. Therefore, a better understanding of MP contamination in sediments is crucial [66].

### 3.1. Sources of Microplastic in Sediments

The main sources of MPs in the sediments are drainage from domestic and industrial wastes, household wastewater, and cosmetic exfoliants [25]. Irfan et al. [52] elucidate some reasons for high MP concentrations in lakes and water bodies, and thus its accumulation in sediments, such as (1) higher population density as the main cause of greater MP amount; (2) the inflow of domestic sewage; (3) proximity to residential and industrial areas, in addition to tourist and aquaculture activities; (4) transporting of MP particles by the land–sea breeze at the water with increasing solar irradiance; and (5) small surface areas of water bodies.

### 3.2. The Impacts of Microplastic on the Sediments

Microplastics either accumulate directly in seabed sediments by sinking through the water column or indirectly by being carried by currents and by sediments that are carried down continental slopes. However, a variety of practical considerations make it difficult to determine the degree of deposition of MPs in deep-sea sediments. In particular, the ocean is deep and wide; therefore, acquiring samples from deep-sea settings is linked to high expenses and practical obstacles. Despite these restrictions, it is crucial to look at MPs in the benthic environment since the deep seafloor may be hiding a lot of plastic waste. MPs have so far been discovered in deep-sea sediments in investigated regions of all significant seas [67]. The fundamental difficulty with the existence of MPs in the aquatic environment is its large surface area and its high affinity for toxins such as dyes, pesticides, heavy metals, medicines, and antibiotics [4]. Low-density particles have a tendency to float on the water's surface or remain suspended in the water column, whereas plastics with

densities greater than seawater's ($1.02 \, \mathrm{g \, cm^{-3}}$) would sink and collect in the sediment. Even low-density polymers can, however, sink to the ocean floor thanks to density alteration. MPs may also sink to the ocean floor because of biofouling by prokaryotes, eukaryotes, and invertebrates, which increases density [23]. However, MPs in water bodies may be diluted due to seasonal variations in water volume and water dynamic behavior. However, dilution hardly happens in the sediment in a static environment, and sediment can easily work as an accumulation environment. Sediments provide also be a source of MPs and habitats for benthic species, which are key parts of food chains [14]. As in the soil, MPs can be ingested by different aquatic organisms, causing physical and psychological harm in these organisms [36]. Microplastics can act as selective niches for fungi and bacteria and change the microbial communities of the water and sediment [43]. Various substances in sediments, including organic carbon, total nitrogen, biogenic opal, carbohydrates, and lignin, have been shown to be influenced by sediment grain size. As grain size decreased, organic carbon increased, as well as mercury and total organic carbon variation. These chemical interactions indicate that the concentration of MPs varies with sediment grain size, indicating that MPs are deposited and accumulate differently in sediment, which has consequences for benthic species and the whole food chain [23].

## 4. Environmental Fate and Risk Assessment of Microplastics

The connection of the cause/effect response is a crucial feature in the research of the repercussions of MPs in the environment. If such experiments are possible using laboratory assays, numerous environmental factors could play a part in the impacts on biota, including the particle itself, the additives and chemical release used during synthesis, plastic background contamination, and the existence of microbial pathogens. Due to the difficulties of identifying all involved chemicals, assessing the risk of environmental "contaminated plastics" is challenging [68]. In soil environments, MPs migrate, partition, and degrade based on their properties (polymer type, shape, size, density), the climate (temperature, rainfall, wind), the physics and biochemistry of the soil (e.g., soil biota), and other environmental conditions (e.g., mechanical disturbances). A negative or positive charge may be present on MP surfaces by oxidation and friction between soil particles, which affects MP properties and migration processes. Soil physical characteristics, soil biota, and agricultural techniques can all have an impact on the vertical and horizontal distributions of MPs in the soil. Surface runoff and/or wind erosion both contribute to the horizontal movement of MPs. It has been discovered that with the rising frequency of wetting–drying cycles, MP migration into the depth increases noticeably [8]. Thus, characterizing and assessing MPs' environmental fate and mobility necessitates an understanding of the impact of numerous environmental processes and routes. When MPs are released from various sources, they form a heterogeneous combination of particles, shapes, and sizes that are released into the environment [69].

However, degradation rates and half-lives of some plastics have been recently estimated in natural environments; for example, low-density polyethylene plastic bags and high-density polyethylene plastic bottles (100 and 500 µm thick, respectively) show half-lives of 3.4 and 58 years, respectively, in marine environments [70]. It is worth mentioning that interactions between microbes and MPs can affect MPs' fate, chemistry, as well as eco-toxicity [54]. After arriving in the soil, the residues of MPs generally degenerate into micro- and nano-plastics and absorb a variety of heavy metals or release organic contaminants into the soil, particularly phthalic acid esters, posing potential risks to soil biology and human health [1]. Previous research showed that commercialized polymers degraded slowly in the soil. It was found that just 0.4 percent of polypropylene (PP) degraded after one year of soil incubation, but no weight loss was detected in the case of polyvinyl chloride (PVC) after 10–35 years of soil incubation. The texture and composition of the soil both play a significant role in the breakdown of synthetic polymers in the soil. It was revealed that clayey soils degraded polymers faster than sandy soils, which might be related to increased soil organic matter (SOM) [5].

The harmful consequences of MPs on humans remain unknown; nonetheless, everyone eats MPs because these particles have been reported in food and drinking water [71]. Furthermore, the leaching of toxic organic additives, which are commonly utilized in the manufacture of plastics, has been identified as a key concern associated with MP contamination [72]. In 2018, 39 experts from the International Council of Chemical Associations stressed the importance of developing an ecological risk assessment system that gives a comprehensive knowledge of the possible implications of MPs. The most commonly used ecological risk assessment methodologies for MPs now employ the pollution load index and pollution hazard index to measure MP pollution levels in different locations, taking MP richness and polymer chemical toxicology into account. The National Research Council of the United States asks that the risk assessment of pollutants include both environmental exposure and biological toxicity. The species sensitivity distribution (SSD) is frequently used to analyze the biological pollutant toxicity, allowing for the risk assessment of pollutants at the levels of the organism's community as well as the ecosystem level. Furthermore, MP pollution is directly linked to human activities, and socioeconomic aspects should be thoroughly included when developing an ecological risk assessment methodology for MPs. Because MPs may migrate to lakes and seas via surface water and atmospheric depositions, MP ecological risk assessment based on multiple environmental media provides a base for future plastic pollution mitigation [73].

## 5. Remediation of Microplastics

Applying remediation strategies to an MP-contaminated environment can change its general characteristics and improve its processes and functions. However, little is known about how remediation may enhance the quality of environments that have been polluted by MPs [13].

### 5.1. Traditional and Emerging Method in Microplastic Remediation

Many traditional chemical or physical methods are used to remediate MPs. Chemical approaches involve the use of chemicals that either act to change or degrade MPs into less complicated forms or produce floc or adhesion, thus removing MPs them by filtering or other procedures. Chemical procedures are those ways in which chemicals are utilized in the treatment/removal of MPs. The essential idea of chemical addition is aggregation, agglomeration, and floc development, which allows MPs to be sedimented or filtered. Coagulation and flocculation are two major methods for MP elimination that have been extensively studied. Coagulation/ flocculation is primarily concerned with the separation of colloidal particles in a solution by neutralizing their charge, forming flocs, and removing them by sedimentation or filtering [31]. Physical and chemical approaches, such as micro/nano-filtering in treatment facilities, are now the more mature technology to remove MPs from the environment. Physical remediation techniques, however, are regarded as inefficient in the cleanup of MPs from contaminated environments. Likewise, the use of synthetic chemicals for remediation in MP-contaminated areas is a less appealing technique because of its complexity, non-green character, and polymer and environmental variability. As a result, given the environmental hazard posed by MPs, there are urgent needs to create cost-effective and ecologically sustainable remediation techniques [11]. However, Arbabi et al. [74] stated that various methods have been applied to remove MPs from the environment, such as replacing microbeds in ocean water with natural materials and using fewer plastic materials. Also, using filtration methods to filtrate the particles between 1 and 5 microns, such as ultrafiltration, membrane, reverse osmosis, and carbon filters that can separate these harmful substances, is also used effectively. Most efficient removal methods for the four dominant MP types (i.e., polyethylene, polystyrene, polypropylene, and polyethylene tetraphthalate) are integrated methods (i.e., physical, physicochemical, chemical, and biological, respectively). When the removal of only polystyrene is considered, the biological methods come first [74]. Badola et al. [31] stated that the adsorption method is effective, but it necessitates extensive study on the size of MPs as well as the adsorbent

substances employed in the process. Magnetic polyoxometalate-supported ionic liquid phase, a physical separation method based on adsorption, is also an effective MP removal technique. In addition to eliminating MPs, it filters organic, inorganic, and microbiological contaminants and may be tuned for a large volume of water at a time. With a 90% success rate, this approach effectively removed polystyrene-type MPs with sizes of 1 and 10 μm. Particles from polyoxometalate-supported ionic liquid phases bond with MPs, which may then be readily extracted using a magnet. Most of the traditional MP removal methods have been used in treating wastewater, such as in Table 1.

**Table 1.** Traditional microplastic removal techniques.

| Media | Remediation or Removal Technique | Efficiency | Ref. |
|---|---|---|---|
| Wastewater | Membrane bioreactor and different technologies such as rapid sand filtration, discfilter, and dissolved air flotation | 99.9% by Membrane bioreactor, 97% by rapid sand filtration, 95 by dissolved air flotation, and 40–98.5 by discfilter | [75] |
| Wastewater | Dynamic membrane supports mesh through filtration | Turbidity 1 NTU after filtration to 20 min verifying the effective removal of MPs | [76] |
| Wastewater | Electrocoagulation | More than 90% | [77] |
| Wastewater | Coagulation and some technologies like ozone, membrane disc-filter, and rapid sand filtration. | Ranged between 75% to 91.9% and increased into >98% after tertiary treatment | [78] |
| Drinking water | Coagulation together with sedimentation and filtration by granular activated carbon | 40.5–54.5% in the first method, 56.8–60.9% in the second method | [79] |
| Drinking water | Coagulation by alum and Al | Water turbidity less than 1.0 NTU (the starting was with turbidity of 16 NTU) | [80] |
| Sea water | Adsorption by fabricated hollow microsubmarines from "hedgehog" magnetic microsubmarine based on sunflower pollen grains. | Removing MPs controllably in a noncontact method | [81] |
| Drinking water and sea water | Series of zirconium metal–organic framework-based foam materials combined with filtration | Up to 95.5 ± 1.2% | [82] |

Borah et al. [83] stated that using a biological catalyst is an emerging green technique for degrading MPs. Plastic particles in soil can be degraded by a variety of bacteria, fungi, and algae [84]. It is widely recognized that only some microbes can breakdown MPs, while some others may be discovered, and only some enzymes play a specialized role in the biodegradation of MP when the polymer type is unknown [11,84]. It has been demonstrated, for example, that the bacterial stain 201-F6 (*Ideonella sakaiensis*) produced enzymes like PETase and MHETase, which hydrolyze PET to an eco-friendly monomer [84]. The chemical compounds of plastic that enable microbial development by providing nutritional sources for them are responsible for its breakdown or decomposition. Microorganisms play an important part in the degradation of plastic materials. For plastic degradation, bacterial (aerobic or anaerobic breakdown) and fungal (anaerobic breakdown) species have been

reported. In comparison to other degrading techniques, microbial degradation of synthetic polymers is a suitable strategy. Examples of the most prominent microbial strains reported for degrading plastic polymers include *Paenibacillus* sp., *Oscillatoria subbrevis*, *Pseudomonas aestusnigri*, *Corynebacterium* sp., *Bacillus* sp., *Aureobasidium pullulans*, *Enterobacter* sp., *Pseudomonas geniculate*, *Aspergillus fumigatus*, *Bacillus niacin*, *Agromyces* sp., *Aspergillus fumigatus*, *Pseudomonas stutzeri*, *Comamonas acidovorans*, *Comamonas acidovorans* TB-35, *Streptomyces badius*, *Arthrobacter paraffineus*, *Microbacterium paraoxydans*, *Micrococcus luteus*, *Phanerochaete chrysosporium*, *Cladosporium sphaerospermum*, and *Penicillium chrysogenum* [85]. Biotechnology is a new bioremediation method that is gaining popularity [11]. Microbial degradation is a practical and cost-effective method of removing MP pollutants [86]. As well, it is an environmentally beneficial approach and merits more investigation to address the MP problems in aquatic and terrestrial ecosystems. Very few studies and reviews have gone into depth into the biological degradation of MPs, but the approach and probable pathways of MP degradation in these ecosystems remain unknown [11]. The microbial breakdown of MPs involves a number of biochemical reaction pathways. The degradation processes of MPs vary depending on their chemical composition and must be properly studied to conserve our environment. Bio-stimulation (the supply of limiting nutrients to stimulate growth of microorganisms) and bio-augmentation (the supply of live cells able to breakdown) are two major mechanisms for increasing the rate of biodegradation of pollutants [86].

Nanotechnology also has recently gained popularity in soil remediation. New novel approaches are also being studied which combine classic cleanup methods with nanobiotechnological approaches. Modern research shows that nanotechnology has a bright future through interaction with other fields. Because of their unique properties, nanoparticles and associated devices and techniques are employed for a wide range of remedial applications. The applications include, but are not limited to, heavy metal pollution treatment, wastewater purification, hydrocarbon remediation, solid waste cleanup, and radioactive material remediation [87]. Thus, nanotechnology might be a viable field to investigate acceptable solutions for cleaning up MPs in the environment. There are a few intriguing ways in which nanotechnology might help with plastic breakdown. The inclusion of nanoparticles in microbial cultures improves plastic biodegradation. For example, the inclusion of $SiO_2$ nanoparticles at various concentrations has been shown to alter the proliferation of plastic-degrading bacteria [86].

*5.2. Biochar Application in Microplastic Remediation*

Harvest agricultural crops produces a big amount of waste in the environment. For example, oilseed rape, which is planted for seed oil production, produces approximately 70% of the above-ground biomass which remains as straw after harvesting the seeds. Also, softwood is a plentifully available biomass in forests, in the form of wood chips, which results from timber production. Thus, the conversion of these waste biomasses and animal wastes also into BC would be helpful to overcome the challenge of this waste handling and at the same time using the produced BC in many environmental applications [13,88]. Biochar-based remediation has shown promise for a variety of contaminants, including persistent organic pollutants, heavy metals, and other sources of toxins. The distinctive characteristics of BC, such as its expanded surface area, enhanced porosity, enrichment in certain types of functional groups, and eco-friendliness, make it the ideal remedy for the removal of various environmental toxins [89]. Because of their tiny size, these particles frequently have a significant surface area and a high capacity for sorbing organic and inorganic pollutants (and potentially toxic metals, pesticides, hormones, pharmaceuticals, etc.), which may have an impact on the persistence, bioaccumulation, toxicity, and mobility of the pollutants in the environment. The importance of particle size in influencing a material's qualities such as specific surface area and sorption capacity cannot be overstated [90–92].

Guo et al. [93] and Zhao et al. [94] stated that as a developing carbonaceous substance, BC is produced primarily from the biomass of different raw materials, such as straw,

defoliation, and animal manure, at low temperatures (less than 700 °C) and in the absence of oxygen. The primary sources of feedstock are agriculture, the food industry, and agricultural biowaste. The most often used materials include rice husks, bagasse, sludge, feces, distiller grains, press cakes from the oil and juice sector, wood chips and pellets, tree cuttings, and crop waste and straw. However, in addition to lignocellulose matter, manufacturing can also be based on biomass sources including sewage sludge, poultry litter, dung, bones, dairy manure, etc. [88,90].

The primary cause of the remarkable stability of BC is the nature of carbon structures because it is a C-rich material. It consists of minerals and many functional groups, enabling it to adsorb different types of contaminants [88,91,92]. In comparison to other aromatic structures of soil organic matter, such lignin, BC has a substantially larger amount of aromatic C and condensed aromatic structures. Amorphous C, which predominates at lower pyrolysis temperatures, turbostratic C (produced at higher temperatures), and graphite C are three different types of condensed aromatic structures that may be found in BCs. High total and organic carbon content, as well as ideal concentrations of micro- and macro-elements (potassium, sodium, magnesium, calcium, copper, zinc, iron, etc.) are all characteristics of BC that make it highly biodegradable [88]. Due to its biodegradation, dispersion, and bioaccumulation, BC has received a lot of interest as a surface sorbent in recent years. BC can be utilized as an adsorbent for immobilizing organic and inorganic contaminants since they are abundant in functional groups that include oxygen and inorganic elements (such as minerals). Temperature, size, and replacement doping with accidental atoms all have a significant impact on the physical and chemical characteristics of BC, which determine how well it performs in the removal of various impurities. Thus, the pyrolysis conditions play a critical role in the process efficiency, degradation behavior, and cost-efficiency of using BC in MP removal. The biochar's structure and surface area are affected by pyrolysis temperatures, which results in differences in its removal effectiveness. However, BC's weak pore structure and few surface-active sites reduce its ability to absorb substances [93,95,96]. However, Siipola et al. [97] reported that the resulting porosity of BC is dependent on its feedstock characteristics; however, physical activation of char and BC to produce activated carbon (AC) using steam, for example, can reduce the need for chemical substances and following washing processes, minimizing the cost of chemical activation and promoting the creation of larger pores in AC. The sorption effectiveness is related to the surface functionality of the adsorbent C, such as a relatively considerable number of O groups, which enhances the sorption of cationic contaminants. However, the number of surface functional groups in physical AC is mostly less than for chemical AC due to the use of higher temperatures. Activated BC has suitable porosity and adequate surface area for tertiary purification of wastewater. Qiu et al. [98] reported that BC generated at relatively high pyrolysis temperatures, in general, may efficiently control organic pollutants by increasing surface area, microporosity, and hydrophobicity. Lower-temperature BC is thought to be more appropriate for eliminating polar organic and inorganic contaminants via O-containing functional groups, electrostatic attraction, and precipitation.

Hanif et al. [4] stated that some advantages of BC in the adsorption or filtration of many pollutants in continuous-flow removal are intriguing, since the preparation procedure is easy and requires no chemicals, and the precursor material is widely available, diverse, and affordable. Nkoh et al. [99] stated that owing to BC's capacity to act as an adsorbent and an immobilizer of contaminants, the use of BC in the remediation of pollutants is more successful. The reactive surfaces of BC contribute to fixing or immobilization. First, the surface of BC has variable charge sites as well as an abundance of O-containing functional groups that interact with the functional groups of contaminants via various methods such as complexation and electrostatic contact. Second, BC's ability to change the soil surface chemistry is critical in immobilizing pollutants and lowering their bioavailability in soil and following uptake by plants. However, the fixation or immobilization of pollutants with BC types can be varied, and they can also be dynamic since BC characteristics change over time. In addition, BC can change the soil properties, and the interaction with soil

components (either mineral or organic components) can also change BC's properties and effect its capacity for removing MPs and other pollutants [100]. Other factors can also affect BC's efficiency in MP removal; Ahmed et al. [101] reported that the retention of MPs by BC is enhanced by increasing the pyrolysis temperature of BC. As well, with time, the retention of MPs by BC is increased, due to entrapping the MPs in the pore spaces of BC or, as reported in Shang and Chi [100], MPs can aggregate with BC by the formation of physical trapping or chemical surface complexes.

When compared to other carbon-based substances, BC is also effective in the removal of MPs from complicated and changeable environmental circumstances from soil as well as water [89,102,103], as in Table 2. The adsorption porosity and surface area are two key elements that must be taken into account for the removal of MPs to be successful, and BC has both characteristics [104]. Badola et al. [31] reported that besides basic sand filters, BC filters also proved positive for MP removal. The BC filters operate on the same basic adsorption and filtering mechanism. The MP retention is due to the large pores of BC filters. The coarse filter surface promotes physical adsorption of MPs within BC particles.

Many authors also stated that BC was an efficient method to remove or mitigate the negative effects of MPs either in water or soil such as Badola et al. [31] and Elbasiouny et al. [29]. Tursi et al. [62] added that MPs may be eliminated at a reasonable cost using BC. Biochar can change the MPs in the medium by sorption and/or suggested microbial biodegradation [83,105,106]. The efficient sorption of MPs and BC with the pollutants may have aided in this separation process [106]. Furthermore, applying BC to MP-polluted soil might enhance soil quality depending on the ingredients of BC and the temperature of pyrolysis [84].

It has been demonstrated that magnetic BC, or iron (II, III) oxide-BC, is efficient in immobilizing MP particles in soil and groundwater [107]. The used magnetic BC also improved the separation of MPs from other pollutants, such as heavy metals, by speeding up the oxidation process [108]. Tong et al. [107] and Kumar et al. [109] also reported BC as a low-cost and efficient adsorbent and its application is carbon-negative and ecologically effective to remediate both inorganic (such as heavy metals) and organic (such as MPs) contaminants in soil and water. Li et al. [110], Tong et al. [107], and Yang et al. [111] stated that the addition of BC or modified BC (ion-modified BC or magnetic BC) to the media affects the plastic retention and the removal of many pollutants, as well their fate in the environment. Tong et al. [107] synthesized BC and magnetic BC ($Fe_3O_4$-BC) through a superficial precipitation at room temperature. They compared the significance of BC and $Fe_3O_4$-BC applications in the deposition and transport behavior of MPs through the breakthrough curve and retained profile of MPs in quartz sand with and without BCs. Their results revealed that the addition of BC and $Fe_3O_4$-BC decreased the transport of MPs and increased their retention in porous media. $Fe_3O_4$-BC more effectively inhibited MP transport than BC. Thus, the addition of BC/$Fe_3O_4$-BC might change the suspension property and increase the adsorption capacity of porous media. This may be attributed to the increased roughness of porous media surface and the negative decrease in zeta potentials of porous media, which contributes to the increasing deposition of MPs. Furthermore, when simulating a rainstorm (by eluting the columns with extremely low ionic strength solution at high flow rate), they revealed that negligible BC and $Fe_3O_4$-BC (<1%) amount were released from the experimental columns after the transport of MPs. Also, a small amount of MPs were detached from the media under these extreme conditions (quartz sand of 16.5%, quartz sand of 14.6% with BC, and quartz sand of 7.5% with $Fe_3O_4$-BC). Hence, their results indicated that magnetic BC can be potentially applied to immobilize MPs in soil or groundwater. The mechanism of action of using magnetic BC in the removal of contaminants may be explained by Yang et al. [111] in their study of the removal of tetracycline hydrochloride (TCH) by magnetic BC, which may be the same for other contaminants, including MPs. They reported that the internal diffusion model revealed that the TCH adsorption by magnetic BC at 700 °C is not only associated with the porous adsorption in BC, but also with other adsorption forces. Furthermore, its remarkable adsorption efficiency

was mostly ascribed to physical (pore filling effect and the pore size distribution, which aids in the diffusion of TCH into the adsorbent's inner surface) and chemical adsorption. The adsorbent's oxygen-containing functional groups and magnetic iron oxides capture TCH through surface complexation, and $\pi$-$\pi$ interaction may facilitate the adsorption. The adsorption trend may also be due to magnetic BC's zeta potential and dissociation constant. Wang et al. [105] studied the efficiency of Mg/Zn-modified magnetic BC (MBC) as adsorbents for MPs (100 mg mL$^{-1}$) in aqueous solution. The removal efficiencies of Mg-MBC and Zn-MBC were 98.75% and 99.46%, respectively. The adsorption process is thought to have occurred due to electrostatic interaction and chemical bonds between MPs and BC. Therefore, they suggested the promising potential application of Mg/Zn-MBCs in MP removal as a low-cost, powerful, and eco-friendly material. As well, Wang et al. [105] stated that the active sites and porous structure of BC are variable and easy to produce, making BC more appealing. Magnetic BC has received a lot of attention for the following reasons. (1) It is simple to obtain by loading magnetic compounds, (2) in a magnetic field, it is simple to separate the magnetic adsorbents, (3) the loaded $Fe_3O_4$ may improve adsorption efficiency, (4) $Fe_3O_4$ is a biocompatible, non-toxic, and recyclable magnetic material. Kumar et al. [7] also outlined the implications of BC with the co-existence of MPs to better understand the coupled effects of both on soil physicochemical characteristics, plant growth, microbial communities, and heavy metals and other toxic substances. The BC was synthesized from different biomasses (i.e., hardwood, corn straw, corncob, pine and spruce bark, and *Prosopis juliflora*) and set batch experiments. The BCs had highly adsorbed MPs (>90%) under variable environmental circumstances, temperature, pH, dose, particle size, and ionic strength, as a result of chemical bonds and electrostatic attraction. Higher temperature promoted higher adsorption in the aqueous solutions; on the other hand, higher dissolved organic matter, nutrients, and pH might show declined adsorption capacity for MPs using BC. They added that BC-amended sand filters showed higher efficiency in removing MPs in column experiments compared to other available biological, physical, and chemical methods. As well, Kumar et al. [7] demonstrated that using BC in saturated column porous media could inhibit various MPs and this is attributable to decreased steric hindrance, electrostatic repulsion, and competitive sorption because of humic acids, cations, and ionic strength. They also recommended more investigation in this field.

**Table 2.** The application of biochar for microplastic removal in soil, water, and sediment.

| Biochar Type | Pyrolysis Temp. | Media | Response | Mechanism | Ref. |
|---|---|---|---|---|---|
| Pine and spruce bark | 475 °C then it was steam activated at 800 °C | Wastewater and storm water | The steam-activated BC were suitable adsorbent for MPs removal | Ion's exchange | [97] |
| Sugarcane bagasse | 350, 550, 750 °C | Water | Nano-plastics removal from BC produced at 750 °C was dramatically greater (>99%), compared to BC-550 (39%) and BC-350 (24%,) | Electrostatic interaction | [89] |
| Sawdust and Mg/Zn modified magnetic BC | 550 °C | Water | Polystyrene removal efficiency ranged from 94.81%, to 99.46%, | Electrostatic and chemical bonding interactions | [105] |
| Peanut shells and $Fe_3O_4$-BC | 500 °C | Sandy porous media | $Fe_3O_4$-BC highly inhibited the transport of polystyrene in porous media (until 92.36% retention efficiency). | Electrostatic adsorption and complexation | [112] |
| Cellulose-BC and $Fe_3O_4$-BC | 400 °C | Porous media (quartz sand) | Decreasing the transport of MPs and increasing their retention in the media | Deposition | [107] |
| Woodchips | 500 °C | Soil | BC could accelerate the removal of MPs | High sorption rate due to increasing soil DOC and larger specific surface area | [113] |

**Table 2.** *Cont.*

| Biochar Type | Pyrolysis Temp. | Media | Response | Mechanism | Ref. |
|---|---|---|---|---|---|
| Oil seed rape and soft wood pellet | 550 and 700 °C | Soil | Soft wood pellet BC enhanced soil enzyme activity and bacterial diversity and evenness compared with oil seed rape in MPs contaminated soil | Microbial activation due to high surface area, high C content and providing essential elements | [114] |
| Rice straw | 700 °C | Water | BC adsorbed 99.96% of MPs | Electrostatic attractions, surface complexation, H-bonding and π-π | [59] |
| Date nuclei | 500 °C | Soil | BC mitigated the negative effect of MPs n soil, plant, and microorganisms | Electrostatic interaction and chemical bonding | [30] |
| Corn straw | 500 °C | Soil | BC amendment enhanced bacterial community species evenness and richness and facilitated N and P metabolism cycle of MPs contaminated soil plants. | BC can promote the balance between roots' nutrient absorption and bacterial community micro-environment in MP contaminated soil. | [115] |
| Palm kernel and coconut shells | 600 °C | Water | Palm kernel shell BC removed higher percentage of MPs (96.65%) than coconut shell. | Filtration or adsorption | [4] |
| Cotton stalk | 650–750 °C | Soil | BC improved shoot dry matter production and significantly alleviated the hazardous effects of MPs. | Promoting microbial activity, enhancing soil nutrients including N, P, and dissolved organic C content | [116] |

By controlling pyrolysis or adding other substances like magnetite, metal nanoparticles, and nano zero-valent iron, BC is activated and changed to improve its characteristics and composition. Nano-biochar (NBC) is also a good waste management solution since it can absorb pollutants and nutrients better than BC can [117]. Additionally, creating BC-based nanocomposites is an approach to creating a new composite material that combines the benefits of BC and other nano-materials while also enhancing the characteristics of BC. It is stated that as compared to micro-sized BC, NBC with a smaller size than 100 nm exhibited higher metal icon mobility in water and soil environments. NBC's ability to act as a carrier may let natural substances and pollutants move more easily than bulk BC, which has the tendency to retain nutrients and immobilize dangerous compounds [95]. However, the utilization of NBC in MP removal, remediation, or negative effect mitigation is rarely studied and needs more investigations in the future.

## 6. Conclusions

The global concern over emerging contaminants and their negative impacts on health and the environment has increased. Anthropogenic activities are seen as the main cause of environmental degradation. Microplastics have also become a major concern, as they are widely distributed in natural ecosystems and originate from various sources. It is addressed as one of the top ten environmental problems, highlighting its impact on biodiversity and human health. Although the production and use of plastics are rapidly increasing and thus its waste, the recovery rate of plastics is very low. Environmentalists have primarily focused on the environmental impacts of microplastics, but there has not been much attention from scientists and the general public. There is a lack of knowledge about MP contamination in agricultural soil and terrestrial ecosystems. Microplastics can be ingested by wildlife and enter the food chain, causing environmental and health effects. They can also absorb heavy metals and organic contaminants from soil solutions, potentially causing problems and loss of ecosystem services. MPs can impact the soil–plant system and soil organisms, but research on MP pollution in soil environments is still in

its early stages. Although microplastics are well-studied in marine environments, more research is also needed in sediments because of a lot of reasons, such as the depth and the difficulty in collecting samples. Various methods, including physical, chemical, and biological approaches, are used to remove MPs from the environment; however, many of their disadvantages are reported. Biochar is an effective and eco-friendly method to remove and adsorb MP particles with many advantages, since the process is simple, requiring no chemicals. Additionally, biochar is readily available and affordable. Therefore, biochar has a successful track record in remediating pollutants due to its ability to act as an adsorbent and immobilize contaminants. The reactive surfaces of BC play a crucial role in immobilization. The variable charge sites and abundance of oxygen-containing functional groups on biochar's surface allow it to interact with contaminants through complexation and electrostatic contact. Furthermore, biochar can alter the chemistry of soil surfaces, reducing the bioavailability of pollutants and their uptake by plants. However, the immobilization of pollutants with different types of biochar can vary, and the characteristics of biochar may change over time, making the process dynamic. Modified biochar is also widely used in removing or mitigating microplastics' negative effects in the environment and nano-biochar may also be a promising method in this process. However, it requires a lot of research.

### 6.1. Limitation and Challenges of Using Biochar in Microplastic Removal from Soil and Sediment

Some limitations and challenges in this regard are considered:

1. Particle size and adsorption efficiency: The surface area and particle size of biochar are two important variables that affect its capacity to adsorb microplastics. Specific biochar characteristics that effectively trap microplastics may be necessary for optimum adsorption efficacy; these characteristics should be carefully taken into account throughout selecting and producing the biochar.
2. Residence period, movement, and mobility: Biochar can remain in the soil or sediment for a considerable amount of time with restricted mobility. It may sink to the bottom of the soil and become buried, minimizing its interaction with microplastics hanging in the top layers. Biochar may occasionally move from its intended place and cause less successful microplastic removal.
3. Heterogeneous distribution of microplastics: The concentrations of microplastics in soils and sediments may spatially and temporally vary and they are not uniformly distributed. It might be difficult to identify hotspots of microplastic pollution, which makes it problematic to apply biochar for successful remediation in the targeted locations.
4. Costs and scalability: One of the biggest obstacles to using and producing biochar on a wide scale is its cost. Significant energy, infrastructural, and resource requirements are involved in the large-scale manufacturing of biochar. Furthermore, the laborious and costly nature of spreading biochar to polluted regions limits its use for large-scale microplastic removal.
5. Environmental hazards and trade-offs: Although biochar is a viable option, it is crucial to take into account any possible trade-offs and risks to the environment related to its use. According to some research, using biochar in some situations may have unexpected ecological effects, such as changing the microbial populations in the soil or the availability of nutrients; these effects should be carefully considered.

### 6.2. Future Research into Biochar and Microplastic Pollution in Soil and Sediment

Biochar has a lot of potential for tackling environmental issues. To improve the effectiveness of biochar for microplastic remediation in soil and sediment, it is imperative to carry out more study and overcome the constraints and difficulties mentioned above. We can pave the path for sustainable management practices and preventive measures by concentrating on the interaction of biochar with microplastics, lowering microplastic mobility, evaluating soil and sediment health, creating better detection tools, and analyzing ecological implications. Interdisciplinary research collaborations can play a critical role

in determining the future of biochar and microplastic research, eventually leading to a healthier and more sustainable world. Thus, more advances in the research into biochar's potential for reducing microplastic contamination and developing sustainable soil and sediment remediation solutions are essential.

**Author Contributions:** H.E. and F.E. have contributed equally to conceptualization, methodology, investigation, data curation, writing—review and editing. All authors have read and agreed to the published version of the manuscript.

**Funding:** This research received no external funding.

**Data Availability Statement:** The data are available when required under the responsibility of the corresponding author.

**Conflicts of Interest:** The authors declare no conflict of interest.

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
