# Peer review of "Addressing the Microplastic Dilemma in Soil and Sediment with Focus on Biochar-Based Remediation Techniques: Review"

_soilsystems, doi:10.3390/soilsystems7040110_

Round 1

Reviewer 1 Report

Comments and Suggestions for Authors

The authors addressed “Addressing the Microplastic Dilemma in Soil and Sediment with Focusing Biochar-based Remediation Techniques: Review’’. The manuscript illuminates an exciting aspect, but some major problems must be addressed. I carefully revised the manuscript and found some major mistakes regarding the effects of microplastics in the soil and its counter effect through the application of Biochar. After careful review, I recommend the major revision to the author and will consider it for publication after careful revision and addressing all the comments.

The abstract section needs to have a stronger justification regarding the current study and also mention clear implications; therefore, I recommend a major revision. I suggest improving the English and removing all the grammatical mistakes. Also, it has some language problems which need to be addressed. There is no future recommendation and concise conclusion which makes the research easily understandable for the readers. Kindly revise it with great care and will consider them for publication after substantial revision. Additionally, use MP or MPs which creates confusion for the readers.

Line19-21: I found some inconsistencies regarding the structure of these sentences. I suggest the authors to revise it.

The introduction section is too long and I suggest to make it shorter. Also, mention the brief three paragraphs introduction which mainly focuses on the current study. Additionally, there are many grammatical mistakes which need to be revised.

I suggest the author remove all the old references and also keep it at 70-80 numbers. Also, I suggest some new references that the authors must incorporate in the current study. These are most recent and relevant work which are very helpful for the authors to read and include in the review article.

·  Finding Microplastics in Soils: A Review of Analytical Methods

·   Impacts of soil microplastics on the crops: A review

· Microplastics meet invasive plants: Unraveling the ecological hazards to agroecosystems

What are the objectives of the current study? What are the hypotheses? It creates a big ambiguity for the readers. Mention the objectives and hypotheses in the current experiment. Therefore, I suggest revising the last paragraph and mentioning them clearly.

Line 37-51: I suggest the author carefully revise these lines and mention clarity in the lines. The author just picks these lines and mentions that authors do this and so on. The authors must represent the flow of the words and maintain quality throughout the text.

Overall, the figures are too simple and I suggest revising them with greater consideration. I suggest the above manuscript read it clearly and generate an idea of the figures from that review article. Also, there are many other review articles that have high-quality figures and must be mentioned for the readers. For reference you can search the below review article.

The development and application of advanced analytical methods in microplastics contamination detection: A critical review

Line 179-183: I think these are not right. Because some MPs did not change the pH of the soil which was proved by various experiments. I suggest carefully revising the manuscript and removing all the ambiguities.

Overall, the manuscript is of low quality and I suggest the authors carefully revise the review article and then submit the revised version.  

Similarly, the topic of the review article mentions the remediation strategy of the microplastics in the soil through biochar, and in the review article you only mention the functions of the MPs in the soil. I carefully read it and learned that first, the author should clear him or herself about the title and then start writing the review article.

I also suggest the author remove and replace old references with new ones. Kindly use up-to-date references from 2020 onwards. Because there are a lot of work on the MPs and its effects on the crops.

Comments on the Quality of English Language

The quality of the language is very bad and needs extensive revision of the manuscript.

Author Response

We would like to express our thanks and appreciations to all reviewers for their constructive comments and their time to review our manuscript. All reviewers' comments are considered and answered carefully.

Reviewer 1

The authors addressed “Addressing the Microplastic Dilemma in Soil and Sediment with Focusing Biochar-based Remediation Techniques: Review’’. The manuscript illuminates an exciting aspect, but some major problems must be addressed. I carefully revised the manuscript and found some major mistakes regarding the effects of microplastics in the soil and its counter effect through the application of Biochar. After careful review, I recommend the major revision to the author and will consider it for publication after careful revision and addressing all the comments.

The abstract section needs to have a stronger justification regarding the current study and also mention clear implications; therefore, I recommend a major revision. I suggest improving the English and removing all the grammatical mistakes. Also, it has some language problems which need to be addressed. There is no future recommendation and concise conclusion which makes the research easily understandable for the readers. Kindly revise it with great care and will consider them for publication after substantial revision. Additionally, use MP or MPs which creates confusion for the readers.

Thanks for your constructive comments and your time for reviewing this manuscript.

  • Abstract: has been changed
  • Grammar mistakes and language problems: Change made through the manuscript.
  • Future recommendations have been added in the conclusion.
  • MP changed through the manuscript into MPs to avoid confusion for the readers.

Line19-21: I found some inconsistencies regarding the structure of these sentences. I suggest the authors to revise it.

Change made and all the abstract is modified.

The introduction section is too long and I suggest to make it shorter. Also, mention the brief three paragraphs introduction which mainly focuses on the current study. Additionally, there are many grammatical mistakes which need to be revised.

Change made and the introduction is revised and summarized, as well grammar are revised

I suggest the author remove all the old references and also keep it at 70-80 numbers. Also, I suggest some new references that the authors must incorporate in the current study. These are most recent and relevant work which are very helpful for the authors to read and include in the review article.

  • Finding Microplastics in Soils: A Review of Analytical Methods
  •  Impacts of soil microplastics on the crops: A review
  • Microplastics meet invasive plants: Unraveling the ecological hazards to agroecosystems

Some references are deleted and both first suggested references are added but the last one is about invasive plants and will be published in 2024 which will make an objection with the editorial board

What are the objectives of the current study? What are the hypotheses? It creates a big ambiguity for the readers. Mention the objectives and hypotheses in the current experiment. Therefore, I suggest revising the last paragraph and mentioning them clearly.

The objective is already mentioned in the last paragraph. We didn’t add hypotheses because this is a review

Line 37-51: I suggest the author carefully revise these lines and mention clarity in the lines. The author just picks these lines and mentions that authors do this and so on. The authors must represent the flow of the words and maintain quality throughout the text.

Change made

Overall, the figures are too simple and I suggest revising them with greater consideration. I suggest the above manuscript read it clearly and generate an idea of the figures from that review article. Also, there are many other review articles that have high-quality figures and must be mentioned for the readers. For reference you can search the below review article.

The development and application of advanced analytical methods in microplastics contamination detection: A critical review

All figures are drawn by authors to simplify the illustrated text for the reader especially as mentioned in the manuscript regarding the gap knowledge, in addition if we figure, specially the suggested reference is published in Elsevier which is not open accessed, this will make a problem with the copyright issue and the price of using the published figure

Line 179-183: I think these are not right. Because some MPs did not change the pH of the soil which was proved by various experiments. I suggest carefully revising the manuscript and removing all the ambiguities.

We added some word to avoid the ambiguities in the lines 195-200

Overall, the manuscript is of low quality and I suggest the authors carefully revise the review article and then submit the revised version.  

Thanks for your comment, we revised the manuscript and did all the best to be in more quality.

Similarly, the topic of the review article mentions the remediation strategy of the microplastics in the soil through biochar, and in the review article you only mention the functions of the MPs in the soil. I carefully read it and learned that first, the author should clear him or herself about the title and then start writing the review article.

Thanks for the comment, this review included initially many issues such as MPs existing in soil and sediment, its sources, negative impacts and fate in the environment, therefore the word dilemma is mentioned in the title to include such of these issues an make the title is brief; then the review started to clarify simply the remediation techniques as an entrance and for the comparison with biochar.

I also suggest the author remove and replace old references with new ones. Kindly use up-to-date references from 2020 onwards. Because there are a lot of work on the MPs and its effects on the crops.

Many references are deleted, and some of the old references are addresses some basic information and included in table 1 which represent examples for the tradition removal techniques of MPs

Reviewer 2 Report

Comments and Suggestions for Authors

I have reviewed your manuscript entitled "Addressing the Microplastic Dilemma in Soil and Sediment with Focusing Biochar-based Remediation Techniques: Review" and would like to offer some constructive feedback and recommendations to enhance the paper before its acceptance. Firstly, the text has some superscript errors, like in Lines 124 and 125, which should be addressed with thorough proofreading.

Additionally, I suggest including a citation to the recent work by “Rahim, H. U., Akbar, W. A., Begum, N., Uddin, M., Qaswar, M., & Khan, N. (2022). Mulches and Microplastic Pollution in the Agroecosystem. In Mulching in Agroecosystems: Plants, Soil & Environment (pp. 315-328). Singapore: Springer Nature Singapore,” in Section 2.1, Lines 141 and 142, to strengthen the foundation of your review.

Regarding Table 1, which discusses the removal of various organic and inorganic pollutants by biochar, it seems somewhat divergent from the primary focus of your review, which is microplastic remediation. Thus, I recommend excluding this section to maintain a more targeted scope.

To enhance the critical discussion of microplastic removal, consider delving deeper into the subject, and addressing limitations, challenges, and controversies.

Finally, it would be beneficial to include a section on future research recommendations to guide future investigations in the field.

These revisions will elevate the quality and relevance of your manuscript, making it a valuable contribution to the study of microplastic pollution remediation by biochar. I look forward to reviewing your revised manuscript and am available for any questions or clarifications. I appreciate your dedication to advancing this area of research.

Comments on the Quality of English Language

Moderate editing of the English language required

Author Response

We would like to express our thanks and appreciations to all reviewers for their constructive comments and their time to review our manuscript. All reviewers' comments are considered and answered carefully.

Reviewer 2

I have reviewed your manuscript entitled "Addressing the Microplastic Dilemma in Soil and Sediment with Focusing Biochar-based Remediation Techniques: Review" and would like to offer some constructive feedback and recommendations to enhance the paper before its acceptance. Firstly, the text has some superscript errors, like in Lines 124 and 125, which should be addressed with thorough proofreading.

 Thanks for your comments and your time to review this manuscript; change made in line 138-139 regarding the superscripting.

Additionally, I suggest including a citation to the recent work by “Rahim, H. U., Akbar, W. A., Begum, N., Uddin, M., Qaswar, M., & Khan, N. (2022). Mulches and Microplastic Pollution in the Agroecosystem. In Mulching in Agroecosystems: Plants, Soil & Environment (pp. 315-328). Singapore: Springer Nature Singapore,” in Section 2.1, Lines 141 and 142, to strengthen the foundation of your review.

We download this chapter but there is no section 2.1 do you mean another section

Regarding Table 1, which discusses the removal of various organic and inorganic pollutants by biochar, it seems somewhat divergent from the primary focus of your review, which is microplastic remediation. Thus, I recommend excluding this section to maintain a more targeted scope.

Thanks for the comment, table 2 is deleted

To enhance the critical discussion of microplastic removal, consider delving deeper into the subject, and addressing limitations, challenges, and controversies.

Thanks so much, change made and some information are added through the manuscript, in addition the limitation and challenges are added in the end of the conclusion section

Finally, it would be beneficial to include a section on future research recommendations to guide future investigations in the field.

Thanks so much, this part is added in the end of review

These revisions will elevate the quality and relevance of your manuscript, making it a valuable contribution to the study of microplastic pollution remediation by biochar. I look forward to reviewing your revised manuscript and am available for any questions or clarifications. I appreciate your dedication to advancing this area of research.

Reviewer 3 Report

Comments and Suggestions for Authors

The work is very interesting and addresses one of the main problems that threaten the environment, as the authors rightly point out. However, I would like to highlight some observations and suggestions.

- The title of the paper, in my opinion, does not fit the content, since about 5% of the text is devoted to a review of remediation techniques based on the use of biochar.

- Figure 2 (L154-155) is not clear, as the authors indicate that the origin of MPs contamination in soil is due to point or non-point origin. This figure should be further clarified as it may lead to confusion.

- The authors use as justification in the text (L242) a quote attributed to Li et al. and classify it as quote number 54, however in the references, quote 54 does not correspond to the authors they refer to (Li et al.).

- The authors use citations of authors, which refer to others. Specifically in L280 (54) they cite Singh & Bhagwat, but these authors refer to Shiklomanov (I.A. Shiklomanov. World freshwater resources. P.H. Gleick (Ed.), Water in Crisis: A Guide to the World's Freshwater Resources, 9780195076288, Oxford University Press, New York (1993), pp. 13-24). This occurs throughout the work on several occasions (L367 and more), resulting in an unorthodox practice that may cause perplexity to the reader and even more to the original authors, since information is assigned to others who are not the originators.

- The text becomes repetitive, so I suggest that the authors make a thorough review and try not to fall into these repetitions.

I consider that the work, being a review on the use of biochar in the remediation of soil contaminated by MPs, should really be presented as such, avoiding repetitions and using bibliographical references correctly.

Author Response

We would like to express our thanks and appreciations to all reviewers for their constructive comments and their time to review our manuscript. All reviewers' comments are considered and answered carefully.

Reviewer 3

The work is very interesting and addresses one of the main problems that threaten the environment, as the authors rightly point out. However, I would like to highlight some observations and suggestions.

- The title of the paper, in my opinion, does not fit the content, since about 5% of the text is devoted to a review of remediation techniques based on the use of biochar.

Thanks so much for this comment, actually the part of remediation is one of the other parts in the review and inside it the focusing was on the biochar, however even that this consider the first review on focusing on the microplastic remediation by biochar, the studies on this regard is on the early stages so we tried to focus on the most important information from the previous studies in this regard. As well some recent references are added.

- Figure 2 (L154-155) is not clear, as the authors indicate that the origin of MPs contamination in soil is due to point or non-point origin. This figure should be further clarified as it may lead to confusion.

Thanks so much: we added a clarification under the figure

- The authors use as justification in the text (L242) a quote attributed to Li et al. and classify it as quote number 54, however in the references, quote 54 does not correspond to the authors they refer to (Li et al.).

Sorry for this mistake, it is 45 not 54 but now changed to 47

- The authors use citations of authors, which refer to others. Specifically in L280 (54) they cite Singh & Bhagwat, but these authors refer to Shiklomanov (I.A. Shiklomanov. World freshwater resources. P.H. Gleick (Ed.), Water in Crisis: A Guide to the World's Freshwater Resources, 9780195076288, Oxford University Press, New York (1993), pp. 13-24). This occurs throughout the work on several occasions (L367 and more), resulting in an unorthodox practice that may cause perplexity to the reader and even more to the original authors, since information is assigned to others who are not the originators.

Thanks for this comment, we really understand and appreciate your comment, however we did this for two reasons; 1) most of the reviewers now asked to delete the old reference and focus on the last five years studies especially this is a review, and 2) if didn’t look to the original references, we did this to be more honest when we cite

- The text becomes repetitive, so I suggest that the authors make a thorough review and try not to fall into these repetitions.

We review it a gain and change made

I consider that the work, being a review on the use of biochar in the remediation of soil contaminated by MPs, should really be presented as such, avoiding repetitions and using bibliographical references correctly.

Thanks so much for all of your comments.

Reviewer 4 Report

Comments and Suggestions for Authors

Dear Authors

thank you for giving me the opportunity to review this paper. In my opinion the basic idea to fulfill the gap regarding the presence of microplastics in sediments and soils is quite acceptable.

However, there are several issues that need to be improved, for example:

Table 1 is inappropriate since it describes the removal application of BC toward various pollutants (organic and inorganic) whereas it is more appropriate to mention the MPs adsorption capacity of different kind of BCs as in Table 3. What about Table 2?

The conclusion section is a simple reiteration of the concepts already expressed in the text lacking personal comments or point of view.

Finally, English is very difficult to understand and there are many grammatical errors. Moreover in some sentences there are words which are inconsistent with the text (such e.g. line 588)

 I believe that this paper can be accepted for publication after a major revision

Comments on the Quality of English Language

Quality of English language is poor and needs to be revised

Author Response

We would like to express our thanks and appreciations to all reviewers for their constructive comments and their time to review our manuscript. All reviewers' comments are considered and answered carefully.

Reviewer 4

Dear Authors

thank you for giving me the opportunity to review this paper. In my opinion the basic idea to fulfill the gap regarding the presence of microplastics in sediments and soils is quite acceptable.

However, there are several issues that need to be improved, for example:

Table 1 is inappropriate since it describes the removal application of BC toward various pollutants (organic and inorganic) whereas it is more appropriate to mention the MPs adsorption capacity of different kind of BCs as in Table 3. What about Table 2?

Thanks for the comment, Table 2 is deleted

The conclusion section is a simple reiteration of the concepts already expressed in the text lacking personal comments or point of view.

Thank you, the conclusion is revised

Finally, English is very difficult to understand and there are many grammatical errors. Moreover in some sentences there are words which are inconsistent with the text (such e.g. line 588)

Thanks for the comment, English grammar is revised and this sentence is deleted “This is the chemical difference between BC and other organic matter that is most noticeable” to avoid the inconsistent with the text (line 617-618)

Round 2

Reviewer 1 Report

Comments and Suggestions for Authors

The authors incorporated all the comments and there are no other comments.

Reviewer 2 Report

Comments and Suggestions for Authors

The authors successfully incorporated all the comments raised by the reviewer. So, I recommend the paper for publication in Soil System.

Comments on the Quality of English Language

Minor English Editing is required. 

Reviewer 4 Report

Comments and Suggestions for Authors

The authors addressed all the raised questions there are no other comments.